# Current Technologies and Uses for Fruit and Vegetable Wastes in a Sustainable System: A Review

**DOI:** 10.3390/foods12101949

**Published:** 2023-05-11

**Authors:** Yingdan Zhu, Yueting Luan, Yingnan Zhao, Jiali Liu, Zhangqun Duan, Roger Ruan

**Affiliations:** 1Institute of Cereal & Oil Science and Technology, Academy of National Food and Strategic Reserves Administration, Beijing 100037, China; zyd@ags.ac.cn (Y.Z.);; 2College of Food Science, Heilongjiang Bayi Agricultural University, Daqing 163319, China; 3Center for Biorefining and Department of Bioproducts and Biosystems Engineering, University of Minnesota, 1390 Eckles Ave., St. Paul, MN 55108, USA

**Keywords:** fruit and vegetable wastes, bioactive compounds, biorefinery, antioxidants, biofuels

## Abstract

The fruit and vegetable industry produces millions of tons of residues, which can cause large economic losses. Fruit and vegetable wastes and by-products contain a large number of bioactive substances with functional ingredients that have antioxidant, antibacterial, and other properties. Current technologies can utilize fruit and vegetable waste and by-products as ingredients, food bioactive compounds, and biofuels. Traditional and commercial utilization in the food industry includes such technologies as microwave-assisted extraction (MAE), supercritical fluid extraction (SFE), ultrasonic-assisted extraction (UAE), and high hydrostatic pressure technique (HHP). Biorefinery methods for converting fruit and vegetable wastes into biofuels, such as anaerobic digestion (AD), fermentation, incineration, pyrolysis and gasification, and hydrothermal carbonization, are described. This study provides strategies for the processing of fruit and vegetable wastes using eco-friendly technologies and lays a foundation for the utilization of fruit and vegetable loss/waste and by-products in a sustainable system.

## 1. Introduction

### 1.1. Global Situation

Owing to imbalances in the growth of the human population and the rapid development of urbanization, natural resources are under severe stress. In particular, the amount of waste has increased significantly. Meanwhile, the United Nations has predicted that the global demand for food may increase by 70%, which would take place mostly in developing countries. The present population of the whole world, 8 billion people, may increase to as much as 9.7 billion by 2050 [1]. The world population is expected to continue to grow by as many as 90 million people annually over the next 30 years [2]. The growing population of developing countries makes it necessary to meet the challenge of adequate food supply. This means we must produce more food to meet human needs with limited global resources (minerals, forests, fertile land, and water). In addition, climate change may cause further stress on human agriculture, as more droughts and floods challenge agricultural productivity all over the globe.

As population growth may be unavoidable, reducing food waste has far-reaching implications for alleviating global resource pressures. According to the Food and Agriculture Organization of the United Nations (FAO), nearly 14% of food produced in the world was lost or wasted post-harvest in 2019 [3]. This survey estimated that if the amount of food lost and wasted were reduced by half, the whole world would still need about 1314 trillion kilocalories per year by 2050 [4]. Food loss and food waste will have many important global consequences [5]. Economically, the waste can reduce the incomes of farmers, as well as increase the expense to consumers. Environmentally, food loss and waste have many negative impacts on the environment, which may include unnecessary emissions of greenhouse gases that cause global warming and lower utilization of land and water resources, which may lead to a reduction in natural ecosystems and their services. According to these data, our global resources continue to be subjected to depletion as a result of efforts to supply food, and this may impact economies and be impacted by global climate change.

The terms “food loss” and “food waste” refer to the edible parts of plants and animals that are removed during production or processing. These losses and wastes are produced during agricultural processing, screening, production, storage, transportation, and consumption. Traditional food loss refers to edible portions of food materials that are spilled, or lost due to spoilage, in storage. The current study’s emphasis on food waste is also related to the quality of food for human consumption and does not necessarily mean that food has been discarded by the consumer; the decision to throw food away can also be taken by food processors and handlers [6]. About 1.3 billion tons of food is lost or wasted every year during production and processing before consumption at the table; one-third of food loss or waste occurs while it is being consumed, which directly results in food loss or waste of about $680 billion in industrialized and developed countries, and about $310 billion in developing countries [7,8]. Data suggest that food loss and waste data in the United States, from farm to table, is higher than 150 million metric tons (MMT), which includes an estimated 70 MMT of edible food each year [9]. Similar data from European households show that vegetable and fruit wastes account for more than 50% of European food waste [10,11]. These data were collected across all European countries and contain both avoidable and unavoidable waste generated as a result of individual behaviors (related to lifestyle, eating habits, and consumption). In particular, the FAO estimate that the highest wastage is from fruits and vegetables, including roots and tubers, at nearly 45–50%. These food wastes included cereals (30%), oilseeds, meat and dairy (20%), and fish (35%) [7]. Figure 1 illustrates global food loss and waste, calculated by both weight and caloric content [5]. A variety of calculations and statistical tests have been performed on these data. However, one result of such analysis is worth emphasizing: there is a greater correlation between greenhouse gas emissions, land utilization, and water consumption with meat waste than with other types of food waste. It is worth noting that although meat waste has a lower calorie loss, it still causes huge economic and environmental losses. Based on the caloric calculation, global cereal food waste accounts for a relatively high proportion of total food waste. In terms of weight, the proportion of global fruit and vegetable loss is relatively high, as shown in Figure 1. These observations are related to the water content of different food materials, the water content of fruits and vegetables being the highest. Even so, we should not waste too many vegetables and fruits. Food loss by individuals from different regions of the world is shown in Figure 2. North America and European countries are the regions with the greatest personal amounts of food waste. North America ranks first in the world, with food waste above 1500 kilocalories per capita/day, followed by Europe, with 748 kilocalories. These total amounts of food waste are 4–5 times more than that of South and Southeast Asia (414 kcal) and Latin America (453 Kcal), the lowest per capita food waste region [5]. More than a third of the food in the United States is wasted annually, and most of this food waste is disposed of in landfills [12]. The larger categories, as estimated by the USDA in 2010, are fruit and vegetables (43.6 billion pounds, BP), dairy products (25.4 BP), and grains (18.5 BP) [13]. Attempts have been made to identify where, from farm to table, food waste occurs. Generally speaking, for developing countries, most of the food waste is a result of the inefficiency of processing, harvesting, transporting, storage, and production equipment, which indicates that food waste generally occurs before reaching the market. In developing countries, fruit and vegetable waste accounts for between 15% and 50% of all fruits and vegetables after harvest [14,15]. In addition, the processing of fruits and vegetables is closely related to geographical location, harvest season, and processing methods [16]. For example, cassava and yam waste levels can reach between 45% and 50% [17,18]. In South and Southeast Asia, papaya waste ranges from 30% to 60% [19]. Because of the ripening characteristics of fruits and vegetables, they account for around 18% to 40% of food losses in India per year due to a lack of effective cold storage during transportation [20]. In contrast, in developed countries such as the United States, the processing stage is much better than that of developing countries.

Among the many factors that have caused the global environmental burden in recent years, the impact of fruit and vegetable waste has been identified as a major problem. For example, the proportion of waste materials produced in most fruit and vegetable processing is usually high [21]. Because of the characteristics of fruits and vegetables, it is natural to generate losses during the process of selecting, washing, cleaning, peeling, and nucleating in the industrialized production and handling of fruits and vegetables. In addition, fruits and vegetables are rich in enzymes and have the characteristics of enzymatic browning. Therefore, waste during processing is inevitable [22]. Due to the high water content of fruits and vegetables, if storage conditions are poor, then they are vulnerable to spoilage, which increases total food loss. Some potential fruit and vegetable wastes across the world are shown in Table 1. According to Table 1, the waste of fruits and vegetables in North America and Europe per capita was twice as much as in South Africa per year [1]. Therefore, it is necessary to vigorously advocate reducing and preventing food waste by improving storage conditions or reducing storage time.

### 1.2. Sustainable Development

The Agenda for Change of Our World: “Sustainable Development 2030” was unanimously adopted in September 2015 by the United Nations [23]. The agenda covers 17 sustainable development goals and 169 specific goals, providing a grand blueprint for global development. The external environment facing developing countries has deteriorated dramatically, and global development is facing serious challenges [24]. The sustainable development agenda of 2030 provides a golden key to global development issues regarding poverty eradication, employment promotion, social protection, and climate change.

The present work aims to review the current literature regarding the utilization and technologies to reduce fruit and vegetable loss/waste and utilize by-products in a sustainable system. This will review potential functional ingredients in feeding nutrition, food processing, bioconversion processes, and current limitations. This can lay a foundation to reduce fruit and vegetable loss/waste and improve the utilization of by-products in a sustainable system in Figure 3.

## 2. Utilization and Technologies to Reduce Fruit and Vegetable Wastes

### 2.1. Fruit and Vegetable Wastes as an Ingredient in Animal Feed and Food Nutrition

If the world population reaches 9.7 billion by 2050, the demand for meat and milk may increase. Animal feed is a growth point that cannot be ignored. Due to the limitations of natural resources such as water and arable land, the best way to ensure a sustainable feed production system is to improve the quality and characteristics of animal feed [25]. Thus, a reduction of waste, expanding feed resources, and their utilization, should be a focus, which will not compete with human foods. In addition, consumers require the production of “clean”, “natural”, and “environmentally friendly/eco-friendly” labeled foods, regardless of the expense [26,27]. Natural ingredients can be added to feed and food, and they must be safe and healthy. Last but not least, additives can also directly or indirectly affect modern food quality, shelf life, and nutritional value [28].

For fruit and vegetable production, around 30% of wastage can be used in the global market as an ingredient for feed and food [29]. For example, nearly 20–30% of grape pomace is a waste by-product from grapes during wine production. The total waste produced during fruit and vegetable juice processing can amount to as much as 30–50%. These waste by-products contain two types of waste: (1) solid waste, such as the peel, skin, and pomace, and (2) liquid waste, such as juice and wash water [30]. Another example, from sugar beet processing, results in 40–70% vegetable pomace and 85% sugar beet residue during processing [31]. These fruit and vegetable wastes and by-products are potential sources of feed additives for animals and food additives for humans and often contain bioactive compounds. These bioactive compounds can be divided into two categories according to the demand for nutrition: (1) vitamins and (2) minerals, which are essential materials for preventing diseases, and maintaining the basic needs of the body. Additionally, non-essential metabolites may be found in food by-products, which can maintain cell metabolism and extend longevity, such as phenolics and carotenoids [32]. The composition of a wide variety of fruit and vegetable wastes and by-products can be quite different. Their specific applications are not limited to the animal feed processing industry but are also closely related to food processing for human consumption. Table 2 shows the efficacy and application of valuable bioactive substances in fruit and vegetable waste in both feed and food processing. The demand for natural antioxidants extracted from low-cost waste and by-products has been summarized.

It is a sustainable system for feed and food circulation using fruit and vegetable wastes, including two parts: (i) the primary loss of fruit and vegetable leaves (olive), the hulls are collected before processing in the fields; (ii) food processing: the pomaces, skins, or residues of fruits and vegetables. For one thing, these wastes can be used directly for animal feed. There are still some limitations to waste as an alternative ingredient in animal nutrition. Yáñez-Ruiz and Molina-Alcaide [73] reported that adding olive cakes caused a heavy burden on the intestine and stomachs of sheep and goats, which is not conducive to growth. Therefore, the extraction technologies and utilization of these fruit and vegetable wastes as nutritional additives become particularly important.

### 2.2. Current Technologies Used for Fruit and Vegetable Wastes in Food Engineering

Fruit and vegetable waste/residues contain a variety of bioactive compounds, as shown in Table 2. However, the extractions of these bioactive compounds may require various processing techniques, and some novel technologies have been applied. These processing techniques are important when attempting to extract these bioactive compounds from fruit and vegetable waste [74]. The methods used to efficiently extract different bioactive substances may depend on the plant parts being processed, such as stems, leaves, peels, or pomace. However, the extraction can be classified into two methods: (1) traditional methods and (2) novel techniques.

A traditional technique may be a classical method that has been used for years. The disadvantages of traditional methods may be due to large solvent consumption and long extraction times, which may lead to high energy consumption. At present, there are several types of extraction solvents that influence sensory characteristics and quality. For example, alcohol is an important solvent, compared to water, because alcohol has a lower boiling point and heat of vaporization, and thus is easier to remove and recover for re-use. Alcohol has already been regarded as “safe” for use as a food additive extraction solvent [75]. To deal with the limitations of traditional extraction methods, emerging technology has been developed. Novel-assisted extraction has the advantages of large extraction capacity and short-time treatment. There are some “green” extraction solvents. Some of these technically assisted extractions are described below.

#### 2.2.1. Microwave-Assisted Extraction (MAE)

MAE uses a magnetic field, electric field, and microwave heating to directly affect polar materials. It transforms into heat through dipole rotation and an ion conduction mechanism. With the increase in pressure and temperature, solute molecules are separated from the sample matrix, and then the solvent is diffused and released. In the study of Pan et al. [76], caffeine and polyphenols extracted from leaves (green tea) by the MAE method had a higher extraction yield and only took 4 min. MAE can be applied for 10 min under acidic conditions (low temperature, moderate pressure) to obtain high molecular weight and moderate viscosity of beet pectin and orange peel [77,78]. In terms of extraction yield, microwave-assisted extraction of antioxidants from mango peel is 1.5–6 times as much as the traditional method [79].

#### 2.2.2. Supercritical Fluid Extraction (SFE)

SFE utilizes its lower viscosity and higher diffusivity, which can diffuse relatively easily in solid materials and improve the extraction yield. The main solvent of SFE is carbon dioxide, which has a relatively low critical temperature and pressure (31.1 °C and 7.4 MPa). On the other hand, carbon dioxide can achieve food-grade purity and safety [80]. Hassas–Roudsari et al. [81] showed that the total phenol content and antioxidant ability of canola seed meal extracted by SFE were the highest at 160 °C. Some studies have shown that SFE can extract lycopene and β-carotene from tomato peel and seeds [82,83]. Apricot pomace, carrot pulp, and tomato (skins, seeds, and tomato paste waste) are extracted by SFE for their β-carotene, polyphenols, and lycopene [80].

#### 2.2.3. Ultrasonic-Assisted Extraction (UAE)

UAE is used to accelerate the release of active ingredients and promote extraction by utilizing the cavitation effect of ultrasound. The high-frequency sound wave of UAE is strong enough, and the cavitation can easily get to the cell walls. UAE can lead to cell expansion and improve extraction efficiency. Various studies have focused on the extraction of bioactive compounds of by-products, such as flavanones hesperidin, catechins, and carotenoids [80,84]. In comparison to conventional, microwave- and ultrasound-assisted extraction methods, intermittent sonication has a higher extraction of pectin from grapefruit [85]. In addition, the extraction of lycopene by combining the MAE–UAE methods is 8% higher than that by a single UAE [86].

#### 2.2.4. Pressurized Liquid Extraction (PLE)

PLE is a method whereby pressure is applied, causing the temperature to be higher than the normal solvent boiling point temperature [87]. The advantage of PLE is that it consumes less time and requires less solvent. The temperature for efficient extraction of procyanidin from red grape pomace by PLE using organic solvent should be controlled above 80 °C [88].

#### 2.2.5. High Hydrostatic Pressure Technique (HHP)

HHP can increase cell permeability and secondary metabolites diffusion through high-pressure cavitation to promote the release of bioactive substances. Xing et al. [33] extracted sulforaphane from raw broccoli by HHP. It showed that the sulforaphane content that was extracted using 10 mM phosphate-buffered saline (PBS) solution at 5000 psi was three times better than the previous extraction yield. According to Guo et al. [89], the extraction of pectin from the peels of pomelo by HHP using ethanol has a high viscosity compared with the traditional extraction method.

#### 2.2.6. Pulsed Electric Field (PEF)

PEF is a potential non-thermal processing technique for food. It induces the critical potential on the cell membranes by an external electric field, which affects pore development, ruptures, and increases cell membrane permeability [90]. PEF can enhance mass transfer and has already been widely used to improve the extraction of phenols, betalains (pigments), and pectin from grape seed, red beetroot, and apple pomace [91,92]. This method could also result in inactive microorganisms and enhance food safety [93].

#### 2.2.7. Enzyme-Assisted Extraction (EAE)

EAE is a pre-treatment by enzyme and is considered an eco-friendly way to extract both bioactive substances and oils. EAE still has some factors (catalyst, molecular size, materials, etc.) worthy of further study. Gaur et al. [94] used EAE to extract oils from mango kernels, soybean, and rice bran. The extraction rates were as follows: 98%, 86%, and 79%. EAE has also made outstanding contributions to the extraction efficiency of lycopene, bay leaves essential oils, and 6-Gingerol, which contain antioxidants, anti-inflammatory and anti-cancer agents, and other biologically active ingredients [95,96,97].

#### 2.2.8. Ionic Liquids Extraction (ILE)

ILE is an extraction method that utilizes an organic liquid as a solvent. These liquids are good ion conductors that result in a high boiling point system and easy solubility [98]. The greatest advantage of ionic liquids as solvents lie in the fact that the extraction process can be done at room temperature to dissolve hydrophilic/hydrophobic molecules [99]. Ionic liquids such as 1-alkyl-3methylimidazolium-based ILs and N, N-dimethylammonium N, and N-dimethyl carbamate have been studied in the extraction of polyphenols and catechin tannins [100,101].

### 2.3. Industry Examples of Utilization in Food Processing

The technologies and utilization of fruit and vegetable by-products have significant potential to assist the whole world. Table 3 illustrates the current study for the utilization of fruit and vegetable by-products in food processing [102,103,104,105]. By-products extracted from fruit and vegetable losses and wastes have great developmental prospects. This application contains significant high-value bioactive suspensions for such by-products as carbohydrates, hemicellulose, and lignin. These bioactive compounds are reintroduced into the food supply chain, resulting in a significant reduction in environmental and economic losses. It can also enhance the acceptance of the flavor and sensory aspects of foods. However, most current studies lack effective in vivo and toxicological experiments to assure the safety and efficacy of a new product.

### 2.4. Current Technologies for Fruit and Vegetable Wastes in Biorefinery

The expanding population of the whole world results in great demand for energy security; about four-fifths of the world’s population relies on fossil fuels [106]. The concept of “reduce, re-use, recycle, and regenerate” has been introduced to promote environmental sustainability [107]. This sustainable system concept will transform the available economic value of raw material from single-type several products to circular-type, and improve the resulting “yield” from the original raw material. This will improve the comprehensive utilization of raw material resources and eliminate the economic, environmental, and social burdens brought about by a single use of resources. Fruit and vegetable waste is a resource from field to folk, which reduces waste in the environment (land, water, and labor). Most fruit and vegetable waste is currently used as animal feed and compost. The rest may be disposed of in landfills, which can cause emissions of methane; unless this methane can be captured in an anaerobic digester and used as renewable natural gas (RNG) to displace fossil natural gas. It is suggested that the carbon footprint of fruit and vegetable wastes may be associated with green house gas (GHG) emissions [108] if they are not otherwise utilized as RNG. Therefore, fruit and vegetable wastes could be utilized in a biorefinery, where biomass-to-energy technologies could convert these wastes into renewable fuels.

The current use of fruit and vegetable waste-to-energy review focuses on the following processes: biological technology, anaerobic digestion (AD) to produce biogas and bio-methane; fermentation to produce bio-ethanol; thermal/thermochemical technologies such as incineration to produce heat and electricity; pyrolysis and gasification to produce syngas and bio-oil/char; hydrothermal to produce carbonization-hydro-char/gas.

#### 2.4.1. Anaerobic Digestion (AD)

AD of fruit and vegetable wastes in landfills produces biogas, bio-methane, and carbon dioxide, with some other gas such as nitrogen, oxygen, and hydrogen sulfide, which may escape and pollute the atmosphere. However, without oxygen, it seems to be a useful circulation on potential biofuels. Considering the high moisture content of fruit and vegetable wastes, together with some biodegradability contents, it includes 75% sugar and hemicellulose, 9% cellulose, and 5% lignin [109]. The AD of fruit and vegetable wastes is carried out through a series of biochemical reactions, where anaerobic microorganisms hydrolyze cellulose, hemicellulose, pectin, and other wastes of fruits and vegetables to form soluble organic matter and then convert them into organic acids, ethanol, hydrogen, and carbon dioxide by acid-producing bacteria, and finally methane by methanogens [110]. The biogas consists of methane (50–70%), which can be used to supply heat for cooking, electrical power generation, and vehicle fuel. Vegetable wastes and by-products present a high methane yield because of hemicellulose. For example, the methane yield of onion waste, potato waste, and carrot pomace are 390 mL, 320 mL, and 198 mL of CH4/g of volatile solid (VS), respectively. The methane yield of fruit waste and by-products may be more variable because of the variety of biomass compositions (peel and pomace are different). The highest is pineapple waste, up to 413 mL of CH4/g of VS, followed by kiwi waste, which is 317 mL of CH4/g of VS [111]. For fruit waste and by-products, the existence of husks could increase lignin concentrations, which can use some pre-treatment. For example, after alkali pre-treatment, the saponification of cellulose and lignin is easy to be biologically hydrolyzed. For vegetable waste and by-products, some have low pH values, which could cause acidification of the AD. For this reason, fruit and vegetable wastes are suitable for co-digestion with some swine feces and urine, which contain high levels of nitrogen and reduce the number of inhibitory acids. The major systems of bioreactors commonly used are the bath, continuous one-stage, and continuous two-stage reactors with methanation apparatus. In Linke’s [112] research, the solid wastes during potato processing for anaerobic biogas production are continuously stirred in a tank reactor at 55 °C. The results showed that with the organic loading rates (OLR) range of 0.8–3.4 gl^−1^ d^−1^, biogas yields showed a reduction from 0.851 to 0.651 g^−1^, and the methane composition was 58–50%. Overall, AD converted fruit and vegetable wastes into methane and carbon dioxide and digested the residues as a conditioner, amendment, and nutrient source for the land.

#### 2.4.2. Fermentation

Bioethanol is usually derived from a feedstock, such as corn or wheat. However, this may result in food competition with human beings. Thus, fruit and vegetable wastes such as banana peels, potato peels, citrus wastes, and cafeteria food wastes have been used to produce bio-ethanol [113] and do not compete with human food. It is also an approach for waste-to-energy circulation. To improve the digestibility of lignocellulose, the production of bio-ethanol is produced by a different method, such as acid, alkali, or enzymatic pre-treatment. Saccharomyces cerevisiae was the culture usually used in fermentation, which can only use hexose sugars. Together with other fermentative organisms, pentose sugars can be used for ethanol processing. The fermentation production of acetone–butanol–ethanol from pineapple peel can be increased significantly by drying pre-treatment [114]. Both AD and fermentation can increase the digestion of fruit and vegetable wastes to energy.

#### 2.4.3. Incineration

Incineration is a relatively mature technique used to convert fruit and vegetable waste via combustion into heat and energy. By incineration, the volume of waste can be directly reduced by up to four-fifths. However, incineration may result in the accumulation of harmful substances, such as dioxins (due to incomplete combustion) and some gases containing heavy metals. The development of the incineration industry, the improvement and monitoring of air emission control systems, and the application of waste materials in heat recovery and power generation have effectively reduced the dependence on fossil fuels. Because of the high moisture content and non-combustible characteristics of fruit and vegetable waste, it is often discarded into the general domestic waste stream and then converted into heat energy by incineration. A positive result came from Korea, which indicated that a dryer-incineration system is a better way to recover energy from organic wastes and support the “waste-to-energy economy” [115]. 

#### 2.4.4. Pyrolysis and Gasification

Pyrolysis is one thermal process with temperature ranges from 400 to 800 °C and converts fruit and vegetable wastes into bio-oil (major) and gas (syngas) in a non-oxygen environment. Gasification partially oxidizes organic wastes at a relatively high-temperature range of 800 to 900 °C to produce a combustible gas mixture [116]. Due to the reaction temperatures of catalytic pyrolysis and gasification, the products vary due to different reactor types. Pyrolysis degrades the fruit and vegetable wastes into char, bio-oil, tar, and syngas [117]. Gasification degrades hydrogenated syngas, which has some advantages by decreasing the dioxins formed from traditional incineration. However, both gasification and pyrolysis work on carbon-based fruit and vegetable wastes. Nutshells (almond shells, pistachio shells, and oil palm shells) have already been investigated in the production of activated carbons by pyrolysis and gasification [118,119,120]. Wang et al. [121] investigated soybean co-pyrolysis using microwave power for maintaining the temperature at 350 °C and indicated that a catalyst could improve the hydrocarbon production of bio-oil. This means that fruit and vegetable wastes are suitable for thermal treatment to produce low calorific value gas, which can be directly used as fuel for gas turbines.

#### 2.4.5. Hydrothermal Carbonization (HTC)

HTC is a thermal process that can deal with high moisture content wastes (80–90%), especially fruit and vegetable wastes. HTC is wet processing that uses a relatively low-temperature range from 180 to 350 °C under pressure and process duration (0.2–120 hours). HTC is now a hot spot for increased attention from many researchers. HTC can convert precipitated fruit and vegetable wastes into bio-energy at relatively low temperatures and treat large amounts of waste in a relatively short time (several hours) without producing odor. Niksiar and Nasernejad [119] studied the adsorption capacity of copper ions in composting water after HTC with rice husk, citrus waste, and olive pomace at different temperatures, which could greatly shorten the reaction time. At the same time, HTC can also recover nutrients from nitrogen-containing liquids and use them as fertilizers, which is conducive to the biological cycle and improves the sustainable economy [108]. The existence of carbon in fruit and vegetable waste, after AD or fermentation, is converted into carbon dioxide and goes into the atmosphere. However, the original carbon of fruit and vegetable wastes remains in the hydrochar product by HTC without gas emissions.

## 3. Final Remarks and Future Directions

Fruit and vegetable waste recycling are necessary for sustainable human development. The current techniques and utilization of fruit and vegetable wastes as biorefinery feedstock have been discussed. The high value-added components, such as sulforaphane, polyphenols, and pectin, using “green” extraction methods, show large extraction capacity and short-time treatment and indicate a great potential for the food industry. For biofuels, anaerobic digestion and hydrothermal carbonization are promising utilization methods to be considered. As opportunities and challenges both coexist, there are still many issues to be studied, such as catalyst types, dosage, and processing conditions. Overall, to reduce and utilize fruit and vegetable waste, the most important issue to be addressed is active public support and participation and continued effort to expand these ideas to viable, industrial-scale renewable businesses.

## Figures and Tables

**Figure 1 foods-12-01949-f001:**
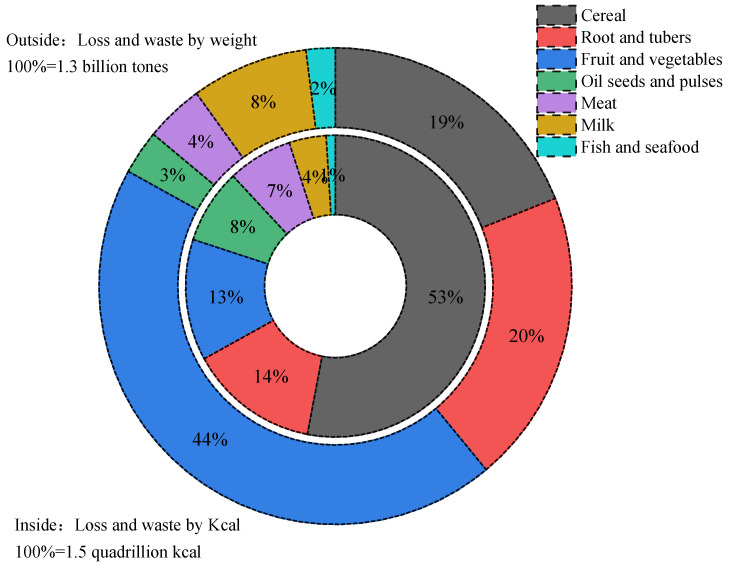
Global food loss and waste by different calculations.

**Figure 2 foods-12-01949-f002:**
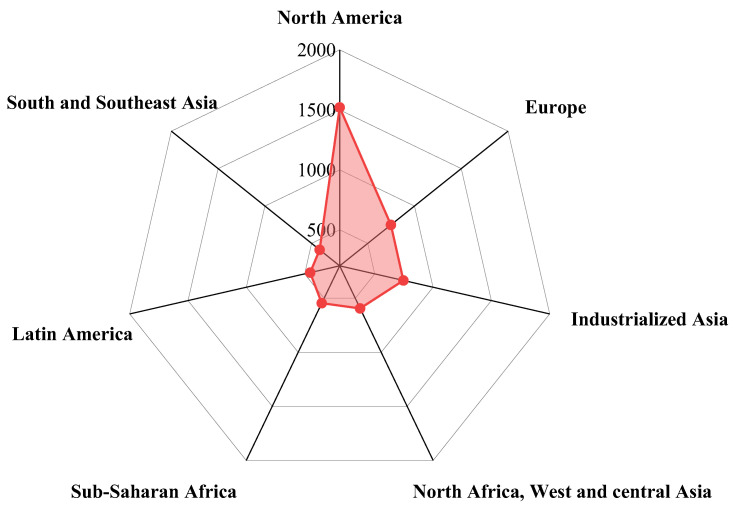
Food loss and waste by region (Kcal/capita/day), according to FAO (2011).

**Figure 3 foods-12-01949-f003:**
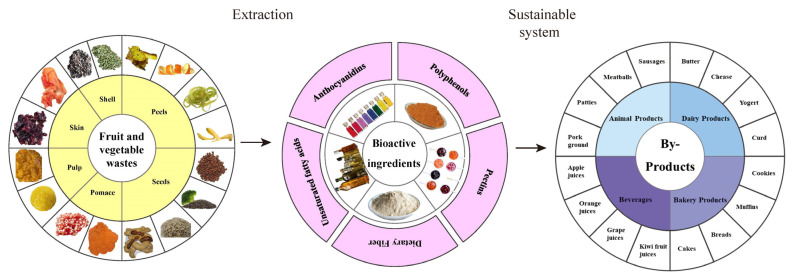
Sustainable system of fruit and vegetable wastes and by-products.

**Table 1 foods-12-01949-t001:** The selected potential fruit and vegetable loss and waste in the world.

Area	Type	Quantity (MT/Year)
North America	Corn stover	80–100 (dry basis)
	Vegetable crop residue	1 (dry basis)
	Tomato pomace	6 × 10^−3^ (California)
	Nutshell and hull	4 × 10^−2^
	Starch	8
Europe	Tomato pomace	4
	Post-manufacture food waste	34
	Citrus waste	0.6 (Spain)
	Olive mill residue	30 (Mediterranean Basin)
	Cocoa pods	20
Africa	Citrus waste	0.14 (South Africa)
	Palm oil residue	15.8 (Indonesia)
Asia	Food waste	1.2 (Hong Kong)
	Citrus residues	9.4
South America-Brazil	Apple pomace	3–4.2
World	Kiwi residue	0.3
	Grape pomace	5–9
	Banana peels	9
	Citrus peel waste	15.6

**Table 2 foods-12-01949-t002:** Bioactive substance ingredients from fruit and vegetable wastes.

Bioactive Substance	By-Products	Function	Reference
Sulforaphane	Broccoli seeds	Against carcinogens and inflammation	[33]
Caffeic acid	Coffee shells	Antioxidant; Anti-bacterial	[34]
Polyphenols	Apple peels	Anti-microbial activity	[35]
	Artichoke		[36]
			[37]
	Cauliflower		[38]
	Date		[39]
	Persimmon by-products		[40]
	Banana peels		[41]
	Mango kernel		[42]
	Orange (peels, pulp)		[43]
	Onion and carrot peel	Antioxidant	[44]
Eugenol	Allspice	Bacteriostatic	[45]
	Alfalfa		[46]
Carotenoids	Tomato (seeds, skins)	Pigment, Radical scavenger	[47]
	Grape pomace		[48]
	Olive		
	Pomegranate pomace		
	Persimmon peels		[49]
	Carrot pomace		[50]
	Guava, orange, and passion fruit by-products		[51]
	Lemon peels		[52]
Flavonoids	Satsuma peels Orange/Lemon (peels, pulp) Banana (peels, roots) Grape (seeds, skin)	Antibacterial, Antioxidant;	[53]
			[54]
		Anti-parasitic, Antioxidant,	[55]
		Food color additives (such as Anthocyanidins)	[56]
			[57]
Pectins	Citrus peels	Thickening agent and emulsification, Food additives	[58]
	Pistachio green hull		[59]
	Pumpkin peels and pulp		[60]
	Tomato wastes		[61]
	Sugar beet (pomace)		[62]
Dietary Fiber	Grapefruit peels, sweet oranges peels, lemon peels	Binders, Texturizers	[63]
	Watermelon rinds, tamarind seed	Low-calorie bulking ingredient	[64]
	Pumpkin by-products		[65]
	Carrot pomace		[66]
	Potato peel		[67]
Unsaturated fatty acids	Tomato seeds	Antioxidant	[68]
	Pistachio pomace		[69]
Saponins	Sapota seeds	Antibacterial	[70]
Amino acids and proteins	Kinnow mandarin waste, pineapple peels, papaya peels,	Protein supplementation	[71]
Glycosides	Banana stem, apple peels	Anti-cancer, Antioxidant	[72]

**Table 3 foods-12-01949-t003:** The current study for application on fruit and vegetable by-products in food processing.

Category	Product Modified	By-Products	Storage Conditions	Key Findings
Animal Products	Beef meatballs, Sausages	Pomegranate peels	8 days (4 °C)	Antioxidant; Anti-bacterial; Antibacterial
			6 months (−18 °C)	
			2 months (−18 °C)	
		Mosambi peels	2 months (−18 °C)	
	Lamb meat, Patties	Tomato pomace Grape pomace Olive pomace Tomato pamace Pomegranate pomace	7 days (2 °C)	Antioxidant; Anti-bacterial
	Chicken meat, Patties, Chickens thigh	Grape pomace, Grapefruit peels, lemon peels Orange and grapefruit peels	14 days (4 °C) 3 months (−18 °C) NA	Antioxidant; Anti-bacteria; Meat qualities and microflora
	Pork ground, Meatballs, Sausages	Persimmon seeds Mango peels Grape seeds	12 days (3 °C) 10 days (4 °C) 12 days (4 °C)	Antioxidant
	Shrimp	Pomegranate peels	10 days (4 °C)	Antioxidant; Anti-bacterial; Meat flavor; Color
	Tuna	Pomegranate peels	10 days (4 °C, 12 °C)	Antibacterial
Dairy Products	Butter	Tomato peel and seeds	2 months (4 °C)	Antioxidant; Anti-bacterial; Flavor; Texture
	Curd	Pomegranate peels	15 days (5 °C)	Antioxidant; Anti-bacterial; Flavor; Texture
	Cheese	Tomato peels and grape pomace	NA	Antioxidant; Texture
	Fermented milk	Grape pomace Olive pomace Grape pomace	28 days (4 °C); 50 days (5 °C)	Antioxidant; Anti-bacterial; Flavor
	Yogurt	Apricot and apple pomace Orange peels Pineapple peels	2 months (−20 °C) 21 days (4 °C) 28 days (4 °C)	Antibacterial; Flavor; Texture; Shorten fermentation time
Beverages	Apple juice	Pomegranate peel	NA	Antioxidant; Antibacterial, Color; Flavor
	Carrot juice	Orange pulp and peels	NA	NA
	Orange juice	Banana peels	30 days (5 °C)	Antioxidant; Color; Flavor
Bakery Products	Cookies	Grape pomace Pineapple central axis Defatted mango kernel	NA	Antioxidant; Color; Flavor, Taste; Texture
	Biscuits	Grape leaves Rowanberry, Blackcurrant, Elderberry pomace	2 months	Antioxidant; Color; Flavor, Taste; Texture
	Bread	Mango peels Pumpkin pomace Grape pomace	NA	Antioxidant; Color; Flavor; Taste; Texture
	Muffins	Raspberry pomace Cranberry pomace Grape peels	NA	Color; Flavor; Taste; Texture
	Cakes	Orange peels Guava seeds Guava pomace Peach palm peels	8 days (25 °C)	Antioxidant; Color; Flavor; Taste; Texture

## Data Availability

No new data were created or analyzed in this study. Data sharing is not applicable to this article.

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
