# Peer review of "Current Technologies and Uses for Fruit and Vegetable Wastes in a Sustainable System: A Review"

_foods, 2023, doi:10.3390/foods12101949_

Round 1
Reviewer 1 Report
In my opinion, this article should be improved since there are some sections which should be removed or corrected.
There are some previous articles which deal with similar topics and have not been included in the references. The authors should carry out a more exhaustive revision.
Trigo et al., (2019). Critical Reviews in Food Science and Nutrition. /doi.org/10.1080/10408398.2019.1572588.
Boye et al., (2013). Food Eng Rev (2013) 5:1–17 DOI 10.1007/s12393-012-9062-z
Esparza et al., ( 2020). Journal of Environmental Management 265 (2020) 110510
Title should be modified since the article is not only focused on the applications of wastes but also on the technologies. Even, this last section (technologies) is more extense than the section of applications, which has been discussed in a very general way.
Section 2. Food loss/waste in the United States. I don’t understand why only the situation in the US is discussed. This section should be dealt in the introduction, giving a world wide view.
Figures 3 should changed or removed since the quality is very low.
Table 2. Antocyanidins are flavonoids. Flavonoids and therefore anthocyanidins are phenolic compounds. This table should be corrected.
Conclusions should be improved.
Author Response
Detailed Response to Reviewer 1 comments
Thank you for your letter and for the reviewers’ comments concerning our manuscript entitled “Fruit and vegetable wastes-potential for feed, food, fuel: A review” (foods-2342636). Those comments are all valuable and very helpful for revising and improving our paper, as well as the important guiding significance to our researches. We have studied comments carefully and have made correction which we hope meet with approval. We also responded point by point to each reviewer comments as listed below, along with a clear indication of the location of the revision.
#Reviewer 1:
Point 1: In my opinion, this article should be improved since there are some sections which should be removed or corrected. There are some previous articles which deal with similar topics and have not been included in the references. The authors should carry out a more exhaustive revision.
Trigo et al., (2019). Critical Reviews in Food Science and Nutrition. /doi.org/10.1080/10408398.2019.1572588.
Boye et al., (2013). Food Eng Rev (2013) 5:1–17 DOI 10.1007/s12393-012-9062-z
Esparza et al., ( 2020). Journal of Environmental Management 265 (2020) 110510
Response 1: Firstly, we sincerely thank you for reading the article and giving us your valuable advice, especially the comments. They are very helpful to improve our paper. We should correct them. Thanks again.
Secondly, we have added the manuscript following the new study above. It can make this study more complete and enriched.
Trigo et al., (2019). Critical Reviews in Food Science and Nutrition. /doi.org/10.1080/10408398.2019.1572588. was in the reference 103. And we have added the last two references in the manuscript.Thank you for your suggestions.
Point 2: Title should be modified since the article is not only focused on the applications of wastes but also on the technologies. Even, this last section (technologies) is more extense than the section of applications, which has been discussed in a very general way.
Response 2: We have modified the title “Current technologies and utilizations for fruit and vegetable wastes from a sustainable system: a review”.
Point 3: Section 2. Food loss/waste in the United States. I don’t understand why only the situation in the US is discussed. This section should be dealt in the introduction, giving a world wide view.
Response 3: We have modified this part and put it in the Introduction section.
Point 4: Figures 3 should changed or removed since the quality is very low.
Response 3: We have rewritten this part and put it in the Introduction section. So we have deleted this part.
Point 5: Table 2. Antocyanidins are flavonoids. Flavonoids and therefore anthocyanidins are phenolic compounds. This table should be corrected.
Response 5: We have rewritten this part. Thanks.
Point 6:Conclusions should be improved.
Response 6: We have rewritten this part. Thanks.
The added reference:
- Matharu, A.S.; de Melo, E.M.; Houghton, J.A. Opportunity for high value-added chemicals from food supply chain wastes. Bioresource technology 2016, 215, 123-130.
- Food; Nations, A.O.o.t.U. Towards the future we want: End hunger and make the transition to sustainable agricultural and food systems. In Proceedings of the Rome: FAO: Policy paper for the Rio+ 20 Conference 13 to 22 June 2012, 2012.
- Esparza, I.; Jiménez-Moreno, N.; Bimbela, F.; Ancín-Azpilicueta, C.; Gandía, L.M. Fruit and vegetable waste management: Conventional and emerging approaches. Journal of Environmental Management 2020, 265, 110510.
- Larrosa, M.; Llorach, R.; Espı́n, J.C.; Tomás-Barberán, F.A. Increase of antioxidant activity of tomato juice upon functionalisation with vegetable byproduct extracts. LWT-Food Science and Technology 2002, 35, 532-542.
- Wang, Y.-C.; Chuang, Y.-C.; Hsu, H.-W. The flavonoid, carotenoid and pectin content in peels of citrus cultivated in Taiwan. Food chemistry 2008, 106, 277-284.
- Ezzat, S.M.; Adel, R.; Abdel-Sattar, E. Pumpkin bio-wastes as source of functional ingredients. In Mediterranean Fruits Bio-wastes: Chemistry, Functionality and Technological Applications; Springer: 2022; pp. 667-696.
- Sengar, A.S.; Rawson, A.; Muthiah, M.; Kalakandan, S.K. Comparison of different ultrasound assisted extraction techniques for pectin from tomato processing waste. Ultrasonics Sonochemistry 2020, 61, 104812.
- Lv, C.; Wang, Y.; Wang, L.-j.; Li, D.; Adhikari, B. Optimization of production yield and functional properties of pectin extracted from sugar beet pulp. Carbohydrate Polymers 2013, 95, 233-240.
- Wedamulla, N.E.; Fan, M.; Choi, Y.-J.; Kim, E.-K. Citrus peel as a renewable bioresource: Transforming waste to food additives. Journal of Functional Foods 2022, 95, 105163.
- Noh, N.A.N.M.; Karim, L.; Omar, S.R. Value-added products from pumpkin wastes: a review. Malaysian Journal of Science Health & Technology 2022, 8, 77-84.
- Chau, C.-F.; Chen, C.-H.; Lee, M.-H. Comparison of the characteristics, functional properties, and in vitro hypoglycemic effects of various carrot insoluble fiber-rich fractions. LWT-Food Science and technology 2004, 37, 155-160.
- Javed, A.; Ahmad, A.; Tahir, A.; Shabbir, U.; Nouman, M.; Hameed, A. Potato peel waste—its nutraceutical, industrial and biotechnological applacations. 2019.

Reviewer 2 Report
The current paper review the literature data regarding the utilization of Fruit and vegetable wastes-potential for feed, food, fuel. Although the paper is interesting, some information is lacking and need improvement.
Some specific comments:
Abstract do not present sufficint information.
Figures should be noted and cited in the text as per indicated in the instructions for authors. Please revise them.
Table 1. Is very messy arranged. Please use the same font and text style. Also please use dots instead of comma.
Moreover in the cited reference in the Table 1 the results presented by the authors are not to be found in this paper. Please revise, or transform the table in the form of text.
Lin, C. S. K., Koutinas, A. A., Stamatelatou, K., Mubofu, E. B., Matharu, A. S., Kopsahelis, N., ... & Luque, R. (2014). Current and future trends in food waste valorization for the production of chemicals, materials and fuels: a global perspective. Biofuels, Bioproducts and Biorefining, 8(5), 686-715.
LN 115-119. How the authors reviewed the literature? Please explain largely how this study was conducted.
Figure 3 A,B and C are unreadable. The font is too small and is very hard to follow the data presented. Not even by zooming in I am not able to understand the legend in Figure 3A. Please revise all the figures and make them clearer.
This type of presentation is not acceptable. In fact, I suggest removing the entire part of subchapter 2. Food loss/waste in the United States since there is only about US food loss. Why not included also Europe, China and so on …
In Table 2, the authors presented the bioactive substance ingredients from fruit and vegetable waste and by-products. In my opinion is not well arranged. First of all, first column should be the by-product, after the column with bioactive substance and function and reference. Moreover under each by-product insert a line to separate them , because is very hard to understand what bioactive compound is attributed to what by-product. Also, make groups of same by-products. For example: citrus peel is the main by-products, but is not very clear if is about oranges, grapefruits, lemon and so on. The cited reference (57) Ciriminna, R., Chavarría‐Hernández, N., Inés Rodríguez Hernández, A., & Pagliaro, M. (2015). Pectin: A new perspective from the biorefinery standpoint. Biofuels, Bioproducts and Biorefining, 9(4), 368-377, is about lemon peel and apple pomace. The second reference about citrus waste (59) Sayas-Barberá, E., Viuda-Martos, M., Fernández-López, F., Pérez-Alvarez, J. A., & Sendra, E. (2012). Combined use of a probiotic culture and citrus fiber in a traditional sausage ‘Longaniza de Pascua’. Food Control, 27(2), 343-350, is about probiotics and orange fiber, but this no not imply that all citrus peels are rich sources of fiber. For example, in this study is it reported that orange and grapefruit waste are rich sources of Vitamin E, Polyphenols, and Antioxidant capacity as bioactive compounds (https://doi.org/10.1080/1828051X.2020.1845576). Other older paper reported that are rich in pectins, carotenoids and other bioactive compounds (https://doi.org/10.1016/j.foodchem.2007.05.086). Please do not make statements only based on two reviewed studies.
Since the paper is about fruit and vegetable waste, I don’t see many vegetable in this paper.
LN 228-305. I don’t see why is important in this paper to talk so much about the extraction methods. If the authors consider that is necessary, I ll suggest to just mention this techniques and which one they consider to be more appropriate and why.
LN 301-304. About what fruit and vegetable by-products the authors consider that are not enough studied? It is unappropriated to make a general statement when in fact is not clear what type of waste is targeted. Moreover, in Table 3 almost all the observations made are about fruits. Few noted exceptions are tomatoes, pumpkin and olive. However, pumpkin waste, is a rich source of antioxidants, omega-3 fatty acids and crude fiber, and was efficiently used in poultry. Olive pomace is also a rich source of omega-3 fatty acids and is very studied in poultry and ruminants to enhance the product quality. Similar reports are available about tomatoes, which are sources of natural pigments for eggs and meat. There are many examples of paper that I can give if necessary.
LN 418 – 430. Here the authors should mention what is the most underutilized fruit of by-products waste and where they have identified this lack, since in the first part of the paper some food loss data are presented.
The entire reference list should be revised as indicated in the journal instructions.
Overall, the main concerns are:
Enhance table 1 and table 2 and explain/comment the differences among the studies.
Revise all the figures presented and make them clearer.
Add data about global food lose and waste.
Introduce more animal studies where these revised vegetable wastes have been studied and what were the findings.
Revise point 3.2. and be more specific to the subchapter title.
Revise point 4.
Revise the reference list.
Author Response
Detailed Response to Reviewer 2 comments
Thank you for your letter and for the reviewers’ comments concerning our manuscript entitled “Fruit and vegetable wastes-potential for feed, food, fuel: A review” (foods-2342636). Those comments are all valuable and very helpful for revising and improving our paper, as well as the important guiding significance to our research. We have studied the comments carefully and have made a correction which we hope meets with approval. We also responded point by point to each reviewer's comments as listed below, along with a clear indication of the location of the revision.
#Reviewer 2:
The current paper reviews the literature data regarding the utilization of Fruit and vegetable wastes-potential for feed, food, fuel. Although the paper is interesting, some information is lacking and need improvement.
Response: Firstly, we sincerely thank you for reading the article and giving us your valuable advice, especially the comments. They are very helpful to improve our paper. We should correct them point by point. Thanks again.
Some specific comments:
Point 1: Abstract do not present sufficint information.
Response 1: Thank you for your advice, we have rewritten the Abstract.
Point 2: Figures should be noted and cited in the text as per indicated in the instructions for authors. Please revise them.
Response 2: Thank you for your advice, we have modified them in the manuscript.
Point 3: Table 1. Is very messy arranged. Please use the same font and text style. Also please use dots instead of comma.
Response 3: We have modified Table 1 in the manuscript and the units of measurement were approved. Thank you for your suggestion.
Point 4: Moreover in the cited reference in the Table 1 the results presented by the authors are not to be found in this paper. Please revise, or transform the table in the form of text.
Lin, C. S. K., Koutinas, A. A., Stamatelatou, K., Mubofu, E. B., Matharu, A. S., Kopsahelis, N., ... & Luque, R. (2014). Current and future trends in food waste valorization for the production of chemicals, materials and fuels: a global perspective. Biofuels, Bioproducts and Biorefining, 8(5), 686-715.
Response 4: We have modified Table 1 in the manuscript. And the reference was added, thanks.
Point 5: LN 115-119. How the authors reviewed the literature? Please explain largely how this study was conducted.
Response 5: The background of this study comes from the author’s study in the United States. Food waste is negative to the environment and may cause greenhouse gases, which could cause global warming directly. Lower utilization of food waste could cause economic losses and climate change. In particular, the FAO estimated the highest food waste was showed from fruit and vegetable loss. For sustainable development, the present work aimed to show the current technologies to make fruit and vegetable wastes in a circular economy.
Line 115-119 was the reference to the comparison of food loss in the United States and European countries. It was total food loss and waste in the world.
Point 6: Figure 3 A,B and C are unreadable. The font is too small and is very hard to follow the data presented. Not even by zooming in I am not able to understand the legend in Figure 3A. Please revise all the figures and make them clearer.
This type of presentation is not acceptable. In fact, I suggest removing the entire part of subchapter 2. Food loss/waste in the United States since there is only about US food loss. Why not included also Europe, China and so on …
Response 6: We have merged the two parts and deleted the Figure 3 to make the United States become part of the whole world. It can help to integrate the logic of the two parts. This suggestion is of great important to us. Thanks.
Point 7: In Table 2, the authors presented the bioactive substance ingredients from fruit and vegetable waste and by-products. In my opinion is not well arranged. First of all, first column should be the by-product, after the column with bioactive substance and function and reference. Moreover under each by-product insert a line to separate them , because is very hard to understand what bioactive compound is attributed to what by-product. Also, make groups of same by-products. For example: citrus peel is the main by-products, but is not very clear if is about oranges, grapefruits, lemon and so on. The cited reference (57) Ciriminna, R., Chavarría‐Hernández, N., Inés Rodríguez Hernández, A., & Pagliaro, M. (2015). Pectin: A new perspective from the biorefinery standpoint. Biofuels, Bioproducts and Biorefining, 9(4), 368-377, is about lemon peel and apple pomace. The second reference about citrus waste (59) Sayas-Barberá, E., Viuda-Martos, M., Fernández-López, F., Pérez-Alvarez, J. A., & Sendra, E. (2012). Combined use of a probiotic culture and citrus fiber in a traditional sausage ‘Longaniza de Pascua’. Food Control, 27(2), 343-350, is about probiotics and orange fiber, but this no not imply that all citrus peels are rich sources of fiber. For example, in this study is it reported that orange and grapefruit waste are rich sources of Vitamin E, Polyphenols, and Antioxidant capacity as bioactive compounds (https://doi.org/10.1080/1828051X.2020.1845576). Other older paper reported that are rich in pectins, carotenoids and other bioactive compounds (https://doi.org/10.1016/j.foodchem.2007.05.086). Please do not make statements only based on two reviewed studies.
Response 7: In this section, the bio-active compounds was emphasized in the paragraph. So the potential bio-active ingredients from fruit and vegetable wastes was necessary on the first column. And we have modified it in the manuscript. Thanks.
Point 8: Since the paper is about fruit and vegetable waste, I don’t see many vegetable in this paper.
Response 8: In the revised Manuscript, the Line 235 the broccoli seeds, Artichoke, Cauliflower, tomato seeds and skins are about the vegetable wastes. In Line 389, the utilization about the onion wastes, potato wastes with anaerobic digestion to make methane as a bio-fuel. Thanks.
Point 9: LN 228-305. I don’t see why is important in this paper to talk so much about the extraction methods. If the authors consider that is necessary, I ll suggest to just mention this techniques and which one they consider to be more appropriate and why.
Response 9: Since we have modified the title of this study “Current technologies and utilizations for fruit and vegetable wastes from a sustainable system: a review”. We have focus on the technologies to utilize these fruit and vegetable by-products. This is the main point of this study, which keep with the theme of this special issue. We do not only focus on the techniques on the bio-active compounds extraction, but also emphasize on the techniques of bio-refinery as bio-fuels.
Point 10: LN 301-304. About what fruit and vegetable by-products the authors consider that are not enough studied? It is unappropriated to make a general statement when in fact is not clear what type of waste is targeted. Moreover, in Table 3 almost all the observations made are about fruits. Few noted exceptions are tomatoes, pumpkin and olive. However, pumpkin waste, is a rich source of antioxidants, omega-3 fatty acids and crude fiber, and was efficiently used in poultry. Olive pomace is also a rich source of omega-3 fatty acids and is very studied in poultry and ruminants to enhance the product quality. Similar reports are available about tomatoes, which are sources of natural pigments for eggs and meat. There are many examples of paper that I can give if necessary.
Response 10: Thank you for your suggestions. Figure 3 was cite from the reference.
Point 11:LN 418 – 430. Here the authors should mention what is the most underutilized fruit of by-products waste and where they have identified this lack, since in the first part of the paper some food loss data are presented.
Response 11: This section is about the utilization of fruit and vegetable wastes as bio-fuels. The anaerobic digestion of fruit and vegetable wastes are carried out through a series of biochemical reactions, where anaerobic microorganisms hydrolyze cellulose, hemicellulose, pectin and other wastes of fruits and vegetables to form soluble organic matter, and then convert them into organic acids, ethanol, hydrogen, and carbon dioxide by acid-producing bacteria, and finally methane by methanogens. It can be used as bio-fuels. This study was not only focus on the fruit and vegetable wastes on food industry, but also focus on the circular economies, such as bio-refinery, bio-fuels. Thank you for your suggestion.
Point 12:The entire reference list should be revised as indicated in the journal instructions.
Response 12: We have modified the reference.
Point 13:Overall, the main concerns are:
Enhance table 1 and table 2 and explain/comment the differences among the studies.
Revise all the figures presented and make them clearer.
Response 13: We have modified the two tables.
Point 14:Introduce more animal studies where these revised vegetable wastes have been studied and what were the findings.
Response 14: We have modified in the manuscript.
Point 15: Revise point 3.2. and be more specific to the subchapter title.
Response 15: We have revised the subchapter title.
Point 16: Revise point 4.
Response 16: We have modified in the manuscript.
Point 17:Revise the reference list.
Response 17: We have modified in the manuscript.
The added reference:
- Matharu, A.S.; de Melo, E.M.; Houghton, J.A. Opportunity for high value-added chemicals from food supply chain wastes. Bioresource technology 2016, 215, 123-130.
- Food; Nations, A.O.o.t.U. Towards the future we want: End hunger and make the transition to sustainable agricultural and food systems. In Proceedings of the Rome: FAO: Policy paper for the Rio+ 20 Conference 13 to 22 June 2012, 2012.
- Esparza, I.; Jiménez-Moreno, N.; Bimbela, F.; Ancín-Azpilicueta, C.; Gandía, L.M. Fruit and vegetable waste management: Conventional and emerging approaches. Journal of Environmental Management 2020, 265, 110510.
- Larrosa, M.; Llorach, R.; Espı́n, J.C.; Tomás-Barberán, F.A. Increase of antioxidant activity of tomato juice upon functionalisation with vegetable byproduct extracts. LWT-Food Science and Technology 2002, 35, 532-542.
- Wang, Y.-C.; Chuang, Y.-C.; Hsu, H.-W. The flavonoid, carotenoid and pectin content in peels of citrus cultivated in Taiwan. Food chemistry 2008, 106, 277-284.
- Ezzat, S.M.; Adel, R.; Abdel-Sattar, E. Pumpkin bio-wastes as source of functional ingredients. In Mediterranean Fruits Bio-wastes: Chemistry, Functionality and Technological Applications; Springer: 2022; pp. 667-696.
- Sengar, A.S.; Rawson, A.; Muthiah, M.; Kalakandan, S.K. Comparison of different ultrasound assisted extraction techniques for pectin from tomato processing waste. Ultrasonics Sonochemistry 2020, 61, 104812.
- Lv, C.; Wang, Y.; Wang, L.-j.; Li, D.; Adhikari, B. Optimization of production yield and functional properties of pectin extracted from sugar beet pulp. Carbohydrate Polymers 2013, 95, 233-240.
- Wedamulla, N.E.; Fan, M.; Choi, Y.-J.; Kim, E.-K. Citrus peel as a renewable bioresource: Transforming waste to food additives. Journal of Functional Foods 2022, 95, 105163.
- Noh, N.A.N.M.; Karim, L.; Omar, S.R. Value-added products from pumpkin wastes: a review. Malaysian Journal of Science Health & Technology 2022, 8, 77-84.
- Chau, C.-F.; Chen, C.-H.; Lee, M.-H. Comparison of the characteristics, functional properties, and in vitro hypoglycemic effects of various carrot insoluble fiber-rich fractions. LWT-Food Science and technology 2004, 37, 155-160.
- Javed, A.; Ahmad, A.; Tahir, A.; Shabbir, U.; Nouman, M.; Hameed, A. Potato peel waste—its nutraceutical, industrial and biotechnological applacations. 2019.

Round 2
Reviewer 1 Report
The article has been improved with the corrections.
Author Response
Detailed Response to Reviewer 1 comments
Thank you for your letter and for the reviewers’ comments concerning our manuscript entitled “Fruit and vegetable wastes-potential for feed, food, fuel: A review” (foods-2342636). We have modified the title “Current technologies and utilizations for fruit and vegetable wastes from a sustainable system: a review”. Thank you for your suggestions.
#Reviewer 1:
Point 1: The article has been improved with the corrections.
Response 1: We sincerely thank you for reading the article and giving us your valuable advice. They are very helpful to improve our study.

Reviewer 2 Report
The authors improved the content of the manuscript.
Please be careful when you submit your paper to follow the instructions for authors.
Figure 3, nicely done, however, is still too small. use the entire length of the page and enlarge the figure.
Please correct in the entire manuscript antioxidant instead of anti-oxidant
Table 2. Dietary fiber section. As I have mentioned in the previous round, not all citrus peels are sources of dietary fibers. Grapefruit peel has the highest content, followed by lime, lemon and so on. My suggestion to avoid confusions add in parantheses what type of peel the authors identified as a source of dietary fiber. Reference 63 is a very interesting review paper, but I have not seen mentioned that all citrus peels are sources of dietary fiber. Please revise.
Enhance table 3. section animal products. citrus peels were very effective on chickens meat quality. Grape pomace extended shelf life in meat. Other by-products increased the overall quality of meat and animals products. Please note that is this table there is no reference.
Good luck!
Author Response
Detailed Response to Reviewer 2 comments
Thank you for your letter and for the reviewers’ comments concerning our manuscript entitled “Fruit and vegetable wastes-potential for feed, food, fuel: A review” (foods-2342636). We have modified the title “Current technologies and utilizations for fruit and vegetable wastes from a sustainable system: a review”. Thank you for your suggestions.
#Reviewer 2:
Point 1: The authors improved the content of the manuscript.
Please be careful when you submit your paper to follow the instructions for authors.
Response 1: Firstly, we sincerely thank you for the valuable advice, and it is helpful for the study. Thank you!
Point 2: Figure 3, nicely done, however, is still too small. use the entire length of the page and enlarge the figure.
Response 2: We have modified the Figure 3. Thanks.
Point 3: Please correct in the entire manuscript antioxidant instead of anti-oxidant
Response 3: We have modified the entire manuscript. Thanks.
Point 4: Table 2. Dietary fiber section. As I have mentioned in the previous round, not all citrus peels are sources of dietary fibers. Grapefruit peel has the highest content, followed by lime, lemon and so on. My suggestion to avoid confusions add in parantheses what type of peel the authors identified as a source of dietary fiber. Reference 63 is a very interesting review paper, but I have not seen mentioned that all citrus peels are sources of dietary fiber. Please revise.
Response 4: We have modified in Table 2. Thank you for the suggestion, the grapefruit peels has the highest content. Thanks.
Point 5:Enhance table 3. section animal products. citrus peels were very effective on chickens meat quality. Grape pomace extended shelf life in meat. Other by-products increased the overall quality of meat and animals products. Please note that is this table there is no reference.
Response 5: We have modified in the Table 3. Thanks.
Good luck!
Thank you for your valuable comments and suggestions for this study!
The added reference:
- Younis, K.; Ahmad, S.; Malik, M.A. Mosambi peel powder incorporation in meat products: Effect on physicochemical properties and shelf life stability. Applied Food Research 2021, 1, 100015.
- Abdel-Naeem, H.H.; Elshebrawy, H.A.; Imre, K.; Morar, A.; Herman, V.; Pașcalău, R.; Sallam, K.I. Antioxidant and antibacterial effect of fruit peel powders in chicken patties. Foods 2022, 11, 301.
105. Vlaicu, P.A.; Untea, A.E.; Panaite, T.D.; Turcu, R.P. Effect of dietary orange and grapefruit peel on growth performance, health status, meat quality and intestinal microflora of broiler chickens. Italian Journal of Animal Science 2020, 19, 1394-1405.
